# Establishment of a Perfusion Process with Antibody-Producing CHO Cells Using a 3D-Printed Microfluidic Spiral Separator with Web-Based Flow Control

**DOI:** 10.3390/bioengineering10060656

**Published:** 2023-05-28

**Authors:** Jana Schellenberg, Michaela Dehne, Ferdinand Lange, Thomas Scheper, Dörte Solle, Janina Bahnemann

**Affiliations:** 1Institute of Technical Chemistry, Leibniz University Hannover, Callinstraße 5, 30167 Hannover, Germany; schellenberg@iftc.uni-hannover.de (J.S.); michaela.dehne@uni-a.de (M.D.); lange@iftc.uni-hannover.de (F.L.); scheper@iftc.uni-hannover.de (T.S.); 2Institute of Physics, University of Augsburg, Universitätsstr. 1, 86159 Augsburg, Germany

**Keywords:** CHO, perfusion, cell retention, monoclonal antibodies, web-based flow monitoring, 3D printing, microfluidic spiral separator

## Abstract

Monoclonal antibodies are increasingly dominating the market for human therapeutic and diagnostic agents. For this reason, continuous methods—such as perfusion processes—are being explored and optimized in an ongoing effort to increase product yields. Unfortunately, many established cell retention devices—such as tangential flow filtration—rely on membranes that are prone to clogging, fouling, and undesirable product retention at high cell densities. To circumvent these problems, in this work, we have developed a 3D-printed microfluidic spiral separator for cell retention, which can readily be adapted and replaced according to process conditions (i.e., a plug-and-play system) due to the fast and flexible 3D printing technique. In addition, this system was also expanded to include automatic flushing, web-based control, and notification via a cellphone application. This set-up constitutes a proof of concept that was successful at inducing a stable process operation at a viable cell concentration of 10–17 × 10^6^ cells/mL in a hybrid mode (with alternating cell retention and cell bleed phases) while significantly reducing both shear stress and channel blockage. In addition to increasing efficiency to nearly 100%, this microfluidic device also improved production conditions by successfully separating dead cells and cell debris and increasing cell viability within the bioreactor.

## 1. Introduction

Fed-batch processes have long been established as the predominant mode of operation for producing monoclonal antibodies (mAb) in mammalian cell cultures on an industrial scale [1]. Recently, however, the focus has increasingly shifted to exploring continuous approaches. Due to ever-increasing industry demand for cost-efficient and resource-conserving production methods that also satisfy the strict requirements of the pharmaceutical market, process intensification began to attract significant industry attention [2]. Perfusion processes have been used in cell culture since the 1990’s, especially for commercial products, such as the recombinant follicle-stimulating hormone (Gonal-f©) and the interferon beta1a (Rebif©) [3,4]. Indeed, these processes have now become the preferred mode of production for labile products due to their continuous removal of old media containing toxic byproducts or inhibitors in exchange for fresh media. Compared to batch or fed-batch processes, perfusion processes can improve product yield and flexibility while simultaneously reducing cost factors such as substantial media requirements and lengthy process duration [4,5,6].

Given a stable operating point, the kinetics inside a perfusion process do not change—including post-translational modifications such as folding and glycosylation [7]. Additionally, due to the continuous removal of supernatant and cell debris, there is no degradation of the product itself (which might otherwise be caused by the accumulation of proteases or other molecules that are often released by dying cells) [8,9]. This means that not only is the cell-specific productivity higher, but homogenous product quality is also better assured while the product still physically remains in the bioreactor. With respect to the perfusion process itself, the method that is used to ensure cell retention is of critical importance. Gravity settlers, continuous centrifuges, or filtration systems are all well-documented within the relevant literature. However, each of these systems has certain advantages and disadvantages—and not all are equally well-suited for use within small-scale bioreactors such as the Ambr250^®^ [10,11,12].

Due to its adaptability to small scale settings, the low-cost production, high separation efficiency, and improved relative reliability benefits that it offers (when compared to other cell retention devices), a 3D-printed microfluidic spiral separator was chosen for use in this system [13,14,15]. The separation of the particles (cells) from the fluid (medium) was achieved by spatially arranging the particles closer to or further away from the inner channel side of the curved channel in the spiral, depending on their size. This stratification effect stems from various lift and drag forces, as well as the so-called Dean forces that shift the equilibrium position of the differently sized particles inside the curved channels (a process known as inertial focusing) [16,17,18]. Due to the underlying forces at work, cells of different sizes can readily be separated from each other, enabling researchers to isolate and recirculate larger (i.e., viable and producing) cells back into the bioreactor. By contrast, smaller (i.e., dead and shrunken) cells, as well as cell debris, can be isolated and removed from the cultivation entirely [16,19,20,21].

Spiral separators have been described frequently in the literature (e.g., for separating blood cells, tumor cells, bacteria, mitochondria, or mammalian cells) [22,23,24,25,26,27,28,29,30], but many of these spirals were produced using soft lithography. Not only is this traditional manufacturing method significantly more complicated and time-consuming than 3D printing, but in many cases, more elaborate 3-dimensional structures can only be achieved by assembling multiple layers. The possibilities of rapid prototyping and the high-resolution offered by 3D printing thus enables much quicker manufacturing and a much finer optimization of high-resolution microfluidic spirals [31,32]. 

In 3D-printing or additive processes, 3-dimensional structures are built up successively, layer by layer. Materials that can be used include UV-curable plastics, thermoplastics, ceramics, graphene-based materials, biomaterials and metal [33,34]. In addition to different materials, there are also different printing techniques to choose from, all of which have certain advantages and disadvantages [35]. For hollow bodies like microfluidic three-dimensional channel structures, high-resolution printing processes are needed. In addition, the materials should be clear, biocompatible and sterilizable to ensure microscopy and sterile use in cell culture without cell death. Inkjet-based techniques such as MultiJet printing with colorless biocompatible UV-curable plastics and a wax support material are particularly suitable for this purpose [36,37]. 

In the MultiJet printing technique, the support or print material is dropped from an array of nozzles. The print material is a light-curing plastic including a photoinitiator that is cured with the aid of UV light. In this way, the 3-dimensional construct is built up drop by drop [36,38]. This technology has the advantage that very low resolution can be achieved, but it has a higher surface roughness compared to other printing technologies such as digital light processing stereolithography (DLP-SLA) [36,39]. After printing, post-processing is necessary to remove the wax support material by heating and thus melting the wax. These post-processing steps are of great importance, as they affect the biocompatibility of the material and thus have a direct influence on its use in cell culture [40,41]. 

For continuous perfusion processes involving mammalian cells, to date, only a few reported studies have used spiral separators for continuous cell retention, and most of them manufactured the spirals using the comparatively inflexible and complex traditional soft lithography method. Furthermore, most perfusions reported in these systems were of relatively short duration (mostly 7–15 days, with only one that was up to 25 days), and production cell lines are rarely used. Similarly, some of the studies did not examine the cell-free fraction during the process [14,15,21,42,43]. In addition, the continuous monitoring of the process is also of great importance for an industrial application for consistent product quality—especially with small reactors, where a minimal deviation from the planned volumes can quickly result in a drastic change to the filling level. So far, only one paper has demonstrated the gravimetric monitoring of the process to control flow rates, and that work did not provide a display of the results, which would have been viewable from anywhere by researchers [42]. 

In this work, we aim to combine the new advantages of fast and easy fabrication of three-dimensional microfluidic spiral separators (using high-resolution 3D printing technology) with the well-established advantages of perfusion processes. In addition, we also focus on the ability of this system to offer researchers non-invasive and continuous process monitoring (which can reduce the cost of production while also increasing both productivity and efficiency) during the entire perfusion process. Accordingly, maximized cell retention and stability, as well as the adjustability of the separation efficiency, were both considered to be key criteria through which we sought to measure the success of our efforts in this regard. 

## 2. Materials and Methods

### 2.1. Cell Lines and Medium

For this study, a CHO DG44 cell line producing an IgG1 monoclonal antibody was used (Sartorius Stedim Cellca GmbH) and cultivated in chemically defined media and feeds of the Sartorius Stedim Cellca Platform, as described by Schellenberg et al. [44]. The commercially available platform includes a stock medium for the seed culture (SMD), as well as production media (PM) for the main culture. Different media blends of perfusion media formulations based on Janoschek et al. [45] were used. Two different feeds were added in for macronutrients such as glucose (feed medium A, FMA) and micronutrients such as amino acids (feed medium B, FMB). The resulting media formulation perfusion media (PFM) contained 91.2% PM, 8% FMA, and 0.8% FMB. For proof-of-concept experiments, 1% of Penicillin/Streptomycin (Sigma Aldrich, St. Louis, MO, USA) was added to the media. 

### 2.2. Seed Culture and Small-Scale Bioreactor Studies

The seed cultures were performed as described by Schellenberg et al. [44]. A small-scale modular bioreactor system (Ambr^®^ 250, Sartorius Stedim Biotech GmbH, Göttingen, Germany) with a working volume of 100–250 mL and online monitoring of dissolved oxygen (DO) and pH was used. The retention device was connected, via bypass, by replacing the septum cap with a metal 4-way addition port (Sartorius Stedim Biotech GmbH). A 4-channel peristaltic pump (Ismatec SA, Glattbrugg, Switzerland) was used to control all additional flow rates for the continuous cultivation. The production culture was inoculated with 1 × 10^6^ cells/mL into 0.24 L production media. All cultivations were performed at agitation of 855 rpm and temperature of 36.8 °C (±0.5 °C) with setpoints for pH at 7.1 (±0.0) and DO at 60% (±20%), respectively. To prevent foaming, a 2% solution of Antifoam C Emulsion (30%, Sigma, Kawasaki, Japan) was added every 12 h. The process lasted 9–22 days, with a 3-day batch phase in the beginning and different continuous phases (i.e., cell retention, cell bleed). 

### 2.3. Offline Analytics

Sampling was performed manually and measured offline. An equivalent flow rate of 6 mL/day of feed media was being added continuously to balance out sampling. Process-specific metabolite concentrations—including glucose, lactate, glutamine, and glutamate, as well as the product IgG1 and lactate dehydrogenase (LDH) activity—were analyzed using a photometric Cedex Bio Analyzer (Roche, Basel, Switzerland). Analysis of the viable cell concentration (VCC) (i.e., the viability as measured by the total cell concentration and the average cell diameter of the cells) was performed using a Trypan Blue Assay-based Cedex HiRes Cellcounter and Analyzer system (Roche). 

### 2.4. Structure and Fabrication of the Microfluidic Device

The general design of the microfluidic spiral separator (including buffer units to enable pulsation-free separation) has already been described by Enders et al. [13]. Previous studies have shown that by utilizing this spiral separator design, cells ranging from 5–20 × 10^6^ cells/mL can be retained with an efficiency of up to 95%. For the present study, however, the spiral design was further optimized, in particular, the channel split and the buffer devices were changed. The size of the buffer device was increased (to 1.15 and 5.37 mL) so that pulsation-free flows could be ensured even at higher flow rates. Furthermore, a second flushing inlet (fi) was added to the inlet buffer device with the separation inlet (si). This allowed the spiral separator to be flushed in order to ensure both process stability and perfect separation during the entire time.

For improved cell retention, the channel split was also changed to inner outlet (io) / outer outlet (oo) 67:33 (previously it had been 50:50). This optimized channel split ratio was then investigated with respect to separation efficiency at different cell concentrations (5, 10, 15 × 10^6^ cells/mL) and flow rates (inlet 1-4 mL/min). Because shear stress continues to increase with increasing flow rate, a flow rate of 1-3 mL/min inside the device is recommended for continuous experiments. The structure of the 3D-printed spiral separator, as well as the buffer device, is shown in Figure 1. 

The fabrication was carried out with the Projet 2500 plus MultiJet 3D-printer (3D Systems, Rock Hill, SC, USA). The used printing and support material was VisiJet^®^M2S-HT90 (3D Systems) and VisiJet^®^M2-SUP (3D Systems). All post processing steps were performed analogously to that reported by Enders et al 2021 [13]. In addition, however, here the assembled device was coated with Detax (DETAX GmbH) to increase transparency for microscopy and rinsed for 1 h with 80 % ethanol and 14 h with ddH_2_O. After connecting all the tubes, the cell retention system was autoclaved at 121 °C for 30 min (Systec VX-150, Systec GmbH, Hesse, Germany).

**Figure 1 bioengineering-10-00656-f001:**
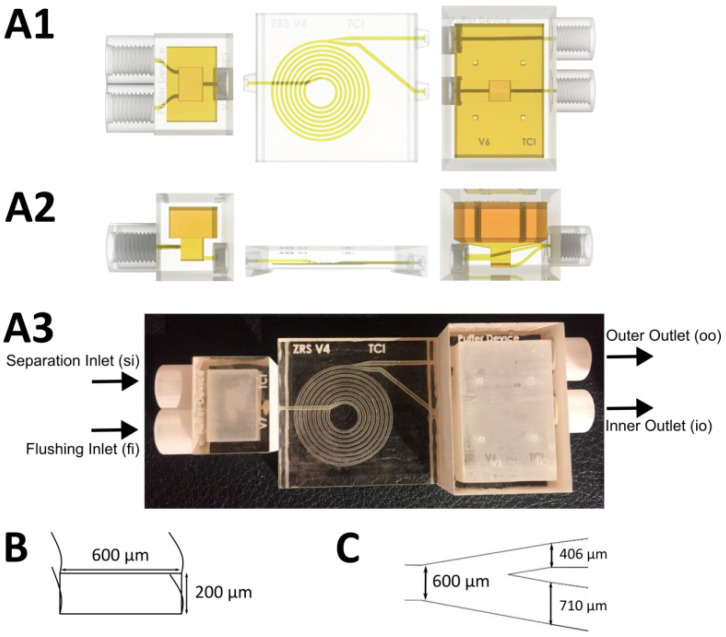
Schematic structure of the 3D-printed spiral separator with buffer devices. A: The 3D structure of the three components, including buffer devices (left and right) and spiral (middle); (**A1**) Top view; (**A2**) Front view; (**A3**) Assembled 3D-printed spiral; (**B**) Channel cross section; and (**C**) Channel split.

### 2.5. Flow Control and Automated Flushing

The entire flow control for separation (si and io) and for automatic flushing (fi) was automatically controlled using a PhythonTM program. The flow rates at the outer outlet (oo) resulted from the hydrostatic pressure difference between the incoming and outgoing flow rates inside the spiral separator. Every 60 min, the cell retention was stopped in order to flush the spiral with a 1:10 diluted PM at a flow rate of 4–14 mL/min for 60–90 s. The cell-free fraction from oo was then collected and gravimetrically analyzed. The gravimetric data of the scale were digitally tapped via an SBI protocol and converted into a flow rate in real time. The gravimetric data together with the time and the calculated flow rate were displayed online in a web-based interface. This web-based interface also offers the possibility of setting a threshold for both maximum and minimum flow rates (see Appendix A). Cell retention phases are characterized by a gradual increase in the weight and step-like drastic increases during automatic flushing (see Appendix A). A mobile alarm is activated if the flow rates deviate from the specified range by a Telegram bot, which then provides status information to multiple users via a group chat [46] (see Appendix A). 

## 3. Results and Discussion

In this work, a perfusion process using a small-scale bioreactor (Ambr^®^ 250) with a 3D-printed microfluidic spiral separator (appropriate for cell retention and web-based flow control) was performed as a proof-of-concept experiment. A Telegram bot was utilized for the web-based flow control and monitoring interfaces. The schematic setup of the experiment is shown in Figure 2, and actual photos of the setup can be found in the Appendix A. Based on the maximum capacity of the spiral separator of 20 × 10^6^ cells/mL, a continuous steady state was targeted at 13–17 × 10^6^ cells/mL. In order to not exceed this limit, a feed and cell-free flow rate at oo was set to 2.5–7.5 mL/h, depending on the growth of the cells.

### 3.1. Implementing Web-Based Flow Control for Real-Time Process Monitoring

For increased process stability during long-term perfusion cultivations, the real-time monitoring of this process is indispensable. Early detection of malfunctions enables rapid correction and constant compliance with process conditions. Online monitoring is usually performed invasively, either in direct contact with the cells in the reactor or in a bypass. Unfortunately, both the residence of the cells outside the optimal conditions of the bioreactor and the mechanical influences of the peristaltic pumps used for bypasses tend to induce undesirable cell stress. 

In this work, the continuous gravimetric determination of the cell-free fraction allowed us to draw direct conclusions about the process stability. In other words, the weight increase in the cell-free fraction (oo) was monitored in real-time to calculate the flow rate. Fluctuations were then minimized via an automatic flushing protocol that was implemented every hour and which was controllable by a web-based flow control monitoring system (see Appendix A). Additionally, a mobile alarm (setup via a Telegram bot) was implemented to notify the user directly on the cellphone if deviant flow rates were detected outside a manually defined range. Here, the transmission of the alarm can be manually started or stopped via both the web-based interface and the Telegram bot. In addition, the Telegram bot was also used to send status queries about the status of the scale, which were always automatically updated to indicate the current flow rate. 

After adjusting the optimal parameters for the web-based interface (e.g., cell-free fraction flow rate at oo 5 mL/h), flow rates (3–6 mL/h), time interval between the measuring points (300 s), and the maximum number of measuring points (500), the system ran very reliably with rare false positive alarm messages. During the hourly pre-programmed flush the web-based alarm mechanism was automatically paused to avoid false positive alarms due to high flow rates. In addition, both the web-based flow monitoring system as well as the Telegram bot offered nuanced control of the process from any place at any time. As there was still no feedback between the web-based interface and the peristaltic pump flow rate changes, additional flushing had to be executed manually in these trials. Accordingly, in future studies, a complete networking of the pump script, as well as the web-based interface, should be prioritized. Furthermore, it was only possible to allow or suppress the alarm from the smartphone in this version, which means that going forward, researchers should attempt to set flow rates directly via the bot. Nevertheless, despite this potential for future exploration and refinement, this web-based flow monitoring worked very well and the constant monitoring of the process via the smartphone proved to be extremely useful.

### 3.2. Results in the Bioreactor

For maximized cell retention efficiency, the 250 mL cultivation began with an increased separation flow rate of 3 mL/min through the spiral separator. The results of VCC and viability, as well as of LDH activity, together with glucose concentration and mAb titer, are summarized in Figure 3. 

The first cell retention phase started at 8 × 10^6^ cells/mL and 98.2% viability after a 3-day batch phase. Afterwards, the internal flow rate of 3 mL/min through the spiral led to a decrease in VCC, along with a significant increase in LDH activity. Because this is indicative of shear stress, the flow rate through the spiral was reduced after 20 h to 2 mL/min. For further reduction in shear stress and increased process stability overnight, the process was switched to a hybrid operation with alternating phases with and without cell retention (152 h). A slight decrease in LDH activity was evident in the cell bleed phases without cell retention.

During the process, the spiral separator was successfully exchanged at both 261 and 454 h without any impairment or contamination of the cultivation, establishing that replacement of the 3D-printed microfluidic system is possible as a plug-and-play system. Expressed differently, as a functional matter, means that design changes can be made and incorporated into the ongoing perfusion process via 3D printing, which is significantly faster and more flexible than traditional production methods. Furthermore, the web-based flow control also successfully facilitated stable cell retention, an exponential VCC increase, and media separation for VCC of up to 17 × 10^6^ cells/mL due to the precise calculations of the cell-free fraction flowrate and the automatic flushing protocol. Moreover, the mAb concentration mirrored the VCC and showed growth-coupled production at optimal glucose concentrations of 3–5 g/L. The process was ultimately stopped after 530 h (22 days) at 13 × 10^6^ cells/mL with a high viability of 97.5% and a mAb titer of 0.3 g/L. 

### 3.3. Results of the Cell-Free Fraction

For a detailed insight into both the efficiency and functionality of the spiral separator, as well as possible product retention, the separated cell-free fraction was collected and analyzed. Figure 4 shows the percentage aggregation rate and the average cell size, as well as the mAb concentration in the reactor and in the cell-free fraction. The aggregation rate is the percentage of cells that are not present individually but in aggregates.

Over the entire course of the process, it can be clearly seen that the aggregation rate within the reactor remained significantly greater than in the cell-free fraction (Figure 4, red). This is because larger particles (i.e., aggregated cells) are retained by the spiral separator. Additionally, the spiral separator mainly separated smaller cells and cell debris, which was confirmed by comparing the average cell size (brown). Generally, the fractionated cells were found to be > 2 µm smaller than in the reactor. The cell aggregation rate increased in the beginning by up to 24% with the use of the first spiral separator, and was then abruptly reduced by 5% due to a spiral exchange at 261 h. Since no increase was observed at later stages of the cultivation, the reduced cell viability at the beginning caused by the higher separation flow rate (3 mL/min) is assumed to have induced cell aggregation. In addition, comparable concentrations of the mAb (orange) in the reactor and in the cell-free fraction indicated that product retention during the continuous separation was negligible. The slight fluctuations and dilutions caused by flushing shortly before sampling were particularly noticeable.

In addition, the VCC and viability of the separated fraction were also deemed to be essential for evaluating retention efficiency. Figure 5A exemplary shows the visible differences in cell morphology as well as cell concentrations and the proportion of dead cells. Here, two things are apparent; the cell count inside the reactor samples are many times higher than in the separated fraction and the fraction contains mostly dead cells and cell debris. Partially viable cells could be detected in the separated fraction due to the flushing; sedimented cells and residues were flushed out of the channels and the pulsation buffers into the cell-free fraction. However, the VCC of max. 0.8 × 10^6^ cells/mL in Figure 5B remained far below the cell concentration within the reactor. The influence of flushing prior to a sample collection was also evident in the widely varying viability (which ranged from 40–100%), although a large proportion of the samples from the cell-free fraction showed cell viability well below 90% (indicated via a gray dashed line), and thus, remained well below the cell viability level observed within the reactor. The significantly higher VCC in the reactor, when compared to the fraction, indicated that cell retention remained almost exclusively at 100%, confirming both the correct operation and the high cell retention efficiency of the separation system.

## 4. Conclusions

In this work, a proof of concept with a 3D-printed spiral separator (incorporating a web-based flow control and automatic flushing system) was successfully utilized as a cell retention system across a long-term perfusion process (22 days). Dead and small cells, as well as cell debris, were all reliably separated in order to keep the cell viability in the reactor above 97%, with cell retention remaining almost exclusively at 100%. Inhibitory by-products—such as lactate—were also continuously removed from the system, thereby preventing the degradation of healthy and productive cells due to toxic accumulation. To keep the separation efficiency constant and to remove possible blocking, which can occur with the rougher surfaces of 3D-printed materials [39], an automatic flushing process and non-invasive flow rate monitoring with web-based control protocol were also deployed. Finally, a mobile alarm system was implemented, which successfully notified the user of any deviations falling outside the manually defined range. 

LDH activity as a shear stress indicator was minimized via a hybrid operation mode of daily cell retention and overnight cell bleed. The implementation of the cell bleed maintained a stable process operation at the targeted cell density, and also reduced overall shear stress. Since no significant product retention was observed, we conclude that the microfluidic cell retention allowed for high product yield without losses or adsorption of the mAb to the channel walls. When used in the context of a recombinant production process, this type of continuous purification (to separate secreted proteases) is essential to maintain high product quality in the separated fraction. The dilution rates of microfluidic approaches, such as periodic countercurrent chromatography (µPCCC), are compatible with the perfusion process in this work [47].

Furthermore, the replaceability of the cell retention system—in case of functional failure or design changes—was also successfully established in this work. The approach that we used to create this system excludes process termination due to irreparable channel blockage, and also facilitates spiral design optimization as a plug-and-play system thanks to the speed and flexibility offered by the 3D printing method. In summary, the spiral separator described in this work offers both a flexible and readily exchangeable alternative to commercially available spiral separators that are made out of stainless steel [48], or spiral separators that are produced using the (more expensive, time-intensive, and inflexible) traditional soft lithography method.

## Figures and Tables

**Figure 2 bioengineering-10-00656-f002:**
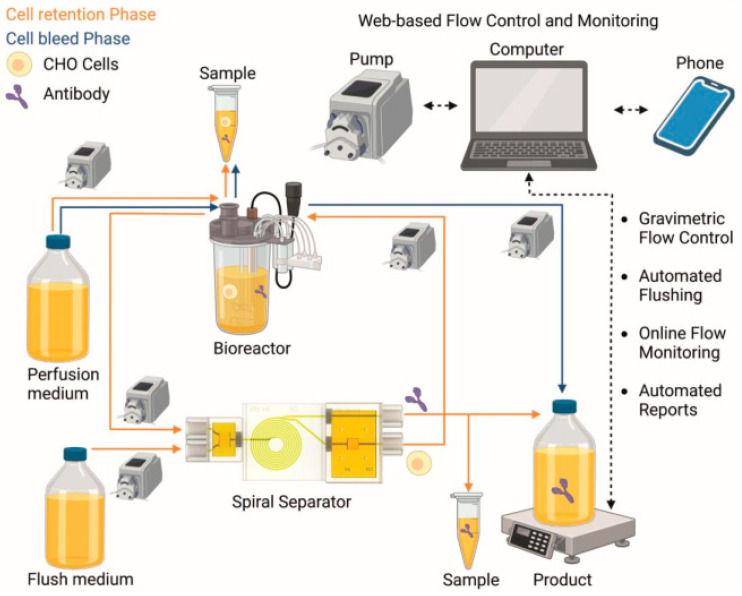
Overview of the entire perfusion process with cell retention phase (orange), cell bleed phase (blue), and web-based flow control and monitoring (black stippled). (Created with Biorender.com).

**Figure 3 bioengineering-10-00656-f003:**
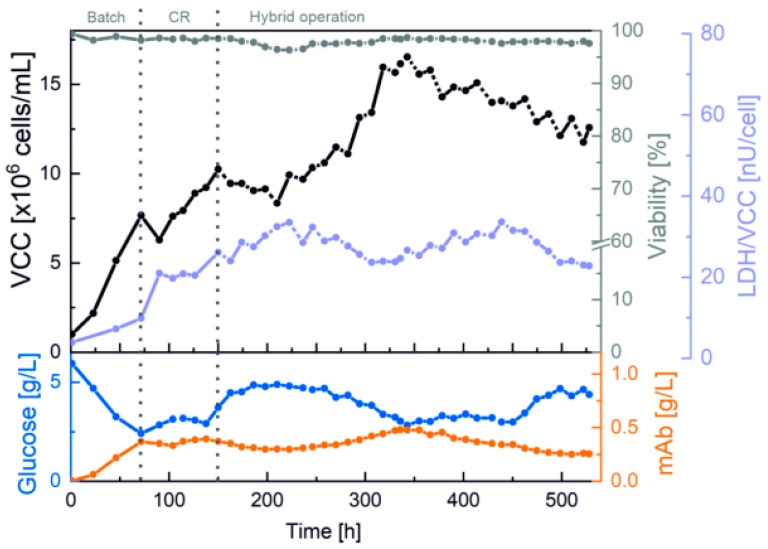
Perfusion cultivation with a 3D-printed spiral separator, automatic flushing and flow rate monitoring in an Ambr^®^ 250 bioreactor system. Offline measurements and cultivation phases.

**Figure 4 bioengineering-10-00656-f004:**
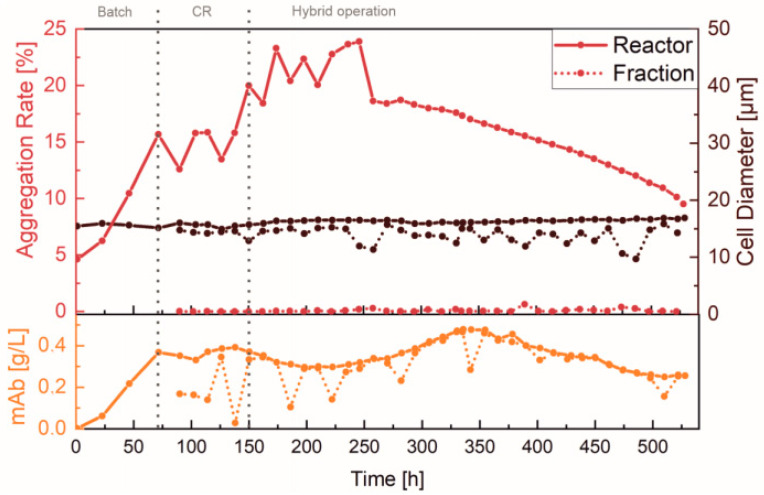
Product retention and cell separation of the perfusion cultivation in an Ambr^®^ 250. Offline measured values of the separated fraction and in the reactor, as well as exchange of the spiral and cultivation phases.

**Figure 5 bioengineering-10-00656-f005:**
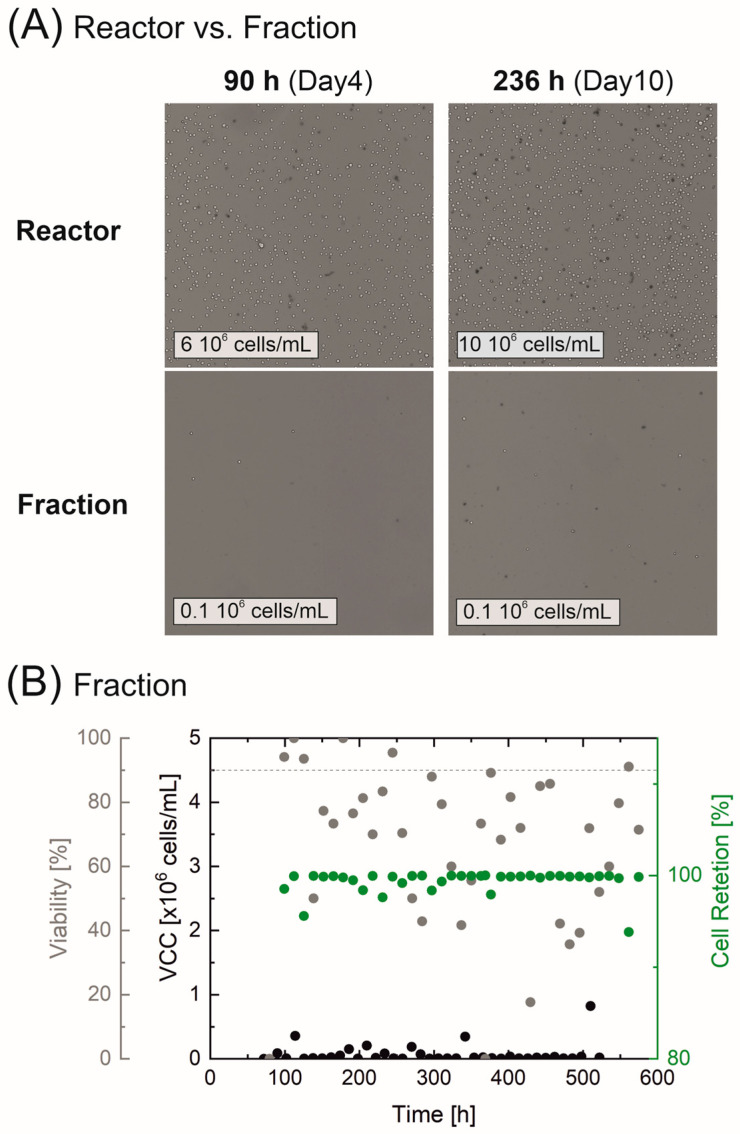
Cell retention of the perfusion cultivation in an Ambr^®^ 250. Microscopic images of the cell morphology (Cedex HiRes: Magnification 10×, dead cells are colored) on exemplary days in the reactor and the separated fraction in (**A**). Cell retention, VCC, and viability of the separated fraction are displayed in (**B**).

## Data Availability

The data presented in this study are openly available in the Research Data Repository of the Leibniz University Hannover at https://doi.org/10.25835/nxrqvp0w (accessed on 26 May 2023).

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
