# Peer review of "Establishment of a Perfusion Process with Antibody-Producing CHO Cells Using a 3D-Printed Microfluidic Spiral Separator with Web-Based Flow Control"

_bioengineering, 2023, doi:10.3390/bioengineering10060656_

Round 1
Reviewer 1 Report
Summary:
This manuscript reports a proof-of-concept study of 3D-printed microfluidic spiral separator for cell retention. The proposed system has many advantages, such as easy to use (plug-and-play), automatic flushing, web-based control, etc. The system can reduce both shear stress and channel blockage and increase efficiency to nearly 100%. The overall manuscript is well-written although it can be further polished.
Major review:
The manuscript can be further improved, for example,
1) Page 7:
Can the authors show some cell images in the reactor if possible? For example, add some successive images or video of the different statuses of cells during the retention, such as different days, or in reactor or fraction.
2) Page 8:
Whether the microfluidic system has been blogged by the aggregated cells? If so, how do the authors solve this problem?
Minor review:
(Didn’t find obvious typos)
Author Response
Dear Editors, dear Reviewers:
Thank you for the opportunity to submit a revised version of our manuscript “Establishment of a Perfusion Process with antibody producing CHO Cells using a 3D-printed microfluidic Spiral Separator with web-based Flow Control”. We very much appreciate the time and effort dedicated to reviewing our manuscript and for your helpful comments.
Below are the reviewers' comments with our responses in blue font. As requested, the changes made in the manuscript are identified via the "Track Changes" feature. The revision was made in consultation with all co-authors.
Reviewer 1
|
|
1) Page 7:
Can the authors show some cell images in the reactor if possible? For example, add some successive images or video of the different statuses of cells during the retention, such as different days, or in reactor or fraction.
Thank you very much for this comment and the great suggestion. We included pictures of our cell counter (CEDEX HiRes) from the reactor and from the separated fraction on exemplary days (4 and 10) in Figure 5A in the revised manuscript. The pictures clearly show how the cell retention works and underline the results from Figure 5B.
2) Page 8:
Whether the microfluidic system has been blogged by the aggregated cells? If so, how do the authors solve this problem?
Thank you for your comment. Due to the retention of larger cells as well as cell aggregates, temporary channel blockage has indeed occurred during the 500 h perfusion cultivation. For clarification, we included an exemplary microscopic image of the blocked channel split in the revised manuscript and provide additional information (please see Supporting Information Figure S4). To prevent malfunction of the retention system we have implemented the automatic flushing of the device. This way, small aggregates can be flushed out precautionarily and temporary blockages can be cleared. In case of persistent blockages, the device can also be flushed manually with flow rates up to 14 mL/min with the flushing media or even be replaced.

Reviewer 2 Report
see attached pdf

Author Response
Dear Editors, dear Reviewers:
Thank you for the opportunity to submit a revised version of our manuscript “Establishment of a Perfusion Process with antibody producing CHO Cells using a 3D-printed microfluidic Spiral Separator with web-based Flow Control”. We very much appreciate the time and effort dedicated to reviewing our manuscript and for your helpful comments.
Below are the reviewers' comments with our responses in blue font. As requested, the changes made in the manuscript are identified via the "Track Changes" feature. The revision was made in consultation with all co-authors.
Reviewer 2:
Point 1: The biggest inconsistency of this article is the fact that authors claim to have
used the 3D-printed microfluidic spiral separator. What is more, they have also
included it in the title of the manuscript. However, they have not described it at all,
they have not written even a single paragraph about 3D printing and its benefits, apart
from writing that it is a “flexible” method. Authors should write down at least two
paragraphs about basic principles and modus operandi of 3d printing methods as well
as one with its benefits and disadvantages.
Point 2: Following point 1, authors can be assisted by including at least the following
references and even more they can find:
10.3844/ajeassp.2023.12.22
• 10.3844/ajeassp.2022.255.263
• 10.1007/s00366-015-0407-0
• 10.1016/j.apmt.2017.02.004
• 10.3390/ijms232314621
• 10.1016/j.promfg.2019.06.089
Response to point 1+2:
Since the focus of our work is on the application of the 3D-printed spiral separator and aims to highlight the great potential of this device for perfusion cultivation (and since we have already described the development and fabrication of this device in detail in other articles) we have not addressed 3D printing as a fabrication method in detail. However, we agree that 3D printing technology is a fascinating method for rapid prototyping and microsystem fabrication in particular, so we have added these aspects to our revised manuscript and cited relevant work (following the references you suggested). We have briefly discussed the principle and advantages and disadvantages of 3D printing in the revised manuscript to fill this gap in our introduction. We hope that we can meet your expectations.
- Point 3: The manuscript version I received for review has no line count, making the review
process very difficult. I suggest the authors to be more careful and use the Journal’s
template.
Thank you for pointing out the errors that probably occurred when uploading the manuscript. We apologize for the inconvenience. Based on the specifications of the manuscript, we have carefully reviewed and revised all formatting once again. In the revised version, the line numbers appear again.
Point 4: What does the statement “fast and flexible 3D-printing” in the abstract mean?? The
material is flexible? The technique is flexible?
Thank you very much for your comment. The statement indeed refers to the 3D printing process, as the freedom for creativity in the design and optimization of the device is much higher due to the almost infinite possibilities of additive manufacturing compared to conventional processes (e.g. soft lithography). This enables rapid prototyping of microfluidic systems and interactive design customization. We have clarified this aspect again by the additional information on 3D printing technology in the introduction of the revised manuscript.
Point 5: What is the material that the separator is made of? What kind of 3d printer was
used for its fabrication? What was the parameter setting??
Thanks for the comment. We have added the essential parameters (type 3D-printer and the materials) in the revised manuscript. The full description of the post processing can be found in the referenced manuscript by Enders et al. 2021 (doi:10.3390/mi12091060).
Point 6: Do not leave blank spaces between paragraphs. This is elementary..
Please excuse the formatting errors that have occurred accidentally. We revised it accordingly (see point 3).

Round 2
Reviewer 2 Report
I want to thank the authors for their revisions and thorough answers.